# Defining Mental and Behavioural Disorders in Genetically Determined Neurodevelopmental Syndromes with Particular Reference to Prader-Willi Syndrome

**DOI:** 10.3390/genes10121025

**Published:** 2019-12-09

**Authors:** Anthony J. Holland, Lucie C.S. Aman, Joyce E. Whittington

**Affiliations:** Department of Psychiatry, University of Cambridge, Cambridge CB2 8AH, UK; ajh1008@cam.ac.uk (A.J.H.);

**Keywords:** genetic syndrome, Prader-Willi syndrome, mental illness, psychosis, major depressive illness, obsessive-compulsive disorder, autism, eating disorder, skin picking

## Abstract

Genetically determined neurodevelopmental syndromes are frequently associated with a particular developmental trajectory, and with a cognitive profile and increased propensity to specific mental and behavioural disorders that are particular to, but not necessarily unique to the syndrome. How should these mental and behavioural disorders best be conceptualised given that similar symptoms are included in the definition of different mental disorders as listed in DSM-5 and ICD-10? In addition, a different conceptual framework, that of applied behavioural analysis, has been used to inform interventions for what are termed ‘challenging behaviours’ in contrast to types of interventions for those conditions meeting diagnostic criteria for a ‘mental disorder’. These syndrome-specific developmental profiles and associated co-morbidities must be a direct or indirect consequence of the genetic abnormality associated with that syndrome, but the genetic loci associated with the syndrome may not be involved in the aetiology of similar symptoms in the general population. This being so, should we expect underlying brain mechanisms and treatments for specific psychopathology in one group to be effective in the other? Using Prader-Willi syndrome as an example, we propose that the conceptual thinking that informed the development of the Research Domain Criteria provides a model for taxonomy of psychiatric and behavioural disorders in genetically determined neurodevelopmental syndromes. This model brings together diagnostic, psychological and developmental approaches with the aim of matching specific behaviours to identifiable neural mechanisms.

## 1. Introduction

Whilst abnormalities at a specific genetic locus can readily be linked to the presence of a specific physical illness (e.g., sickle cell anaemia and Huntington disease) establishing such similarly close relationships between single genes and particular behaviours and/or mental disorders has been much more elusive. The study of neurodevelopmental syndromes in which their genetics is known and the risk of specific behaviours and/or mental disorders are increased has the potential for providing new insights into the genetic influences on the aetiology of such problems. However, in order to investigate the mechanisms that underpin the rates of syndrome specific behavioural and psychiatric disorders it is important to be clear as to how these are best defined in a manner that more directly links them to known neural systems. Similarities in behaviour occur across neurodevelopmental syndromes: for example, intermittent aggressive or self-injurious behaviour. Just as there are likely to be different reasons for why a person has high blood sugar, so there are likely to be different reasons for why people engage in aggressive or self-injurious behaviour. However, there may be a common mechanism for similar behaviours within a specific syndrome. In each case the behaviours may have their own identifiable characteristics in terms of the age of onset, course, and the exact form of the behaviour. In clinical practice it is the diagnostic process and the specificity of the clinical history, physical examination, and investigations that enable the link to be made between reported symptoms or observed signs with the putative underlying pathology, and to treatment. Once the behavioural and psychiatric phenomenology are clearly defined, the question can be asked as to whether such phenomena are a consequence of abnormalities in the brain structure and function or whether the behaviour that is observed is better conceptualized as being primarily shaped by environmental contingencies. New treatments for such behaviours are unlikely to emerge from an understanding of the syndrome genetics alone but instead through an understanding of the unique impact of the syndrome genotype on brain development and neural function and the extent and nature of any interplay between the organic and the psychological.

We consider three conceptual frameworks for the description and/or classification of mental and behavioural disorders. These are: the diagnostic approach, applied behavioural analysis, and Research Domain Criteria. We consider how each approach can add to our understanding of such disorders, and why different approaches evolved. Firstly, dissatisfaction with the traditional DSM-5 [1] and ICD-10 [2] approach to classification and diagnosis (diagnostic approach) in respect of behavioural disorders, often termed ‘challenging behaviours’, has led to the application of applied behavioural analysis (ABA) to the understanding of such behaviours. However, this, in turn, fails to fully explain those behaviours that are prevalent in, and specific to, a given genetic syndrome. Now a new framework (Research Domain Criteria) has been proposed that we argue provides a better framework for understanding ‘challenging behaviour’ in the context of neurodevelopmental syndromes by seeking to match such behaviours to underlying neural mechanisms.

The aims of this paper are to explore these issues using Prader-Willi syndrome (PWS) as an example. We begin with a description of PWS, followed by a section on genetically determined neurodevelopmental syndromes, from which we have chosen PWS as a representative to illustrate our arguments. We then describe the three different models for the classification of mental and behavioural disorders, before showing that each can contribute to understanding and/or treating such disorders in PWS.

## 2. Prader-Willi Syndrome (PWS)

Prader-Willi Syndrome is a rare disorder with a birth incidence rate estimated at 1:25,000 [3,4,5,6] and UK prevalence around 1:50,000 [3]. The underlying cause of PWS is the loss of expression of maternally imprinted (paternally expressed) genes from the q11–q13 region of the paternally inherited chromosome 15. This can occur in one of two ways: part, or whole, of the region is deleted (deletion subtype); or two maternally marked chromosome 15 s, but no paternally marked chromosome, are inherited (maternal disomy subtype). There are two common deleted regions: Type 1 between breakpoints 1 and 3 and Type 2 between breakpoints 2 and 3. Maternally marked chromosomes in disomy subclasses are both inherited from the mother (mUPD) or one from the mother and one from the paternal grandmother (imprinting centre defect (IC defect)). The 15q11-q13 region contains both imprinted and non-imprinted genes (Figure 1), the imprinted genes *SNORD116* (human studies) and *MAGEL2* (mouse studies) being considered central to PWS.

Although the underlying genetic abnormalities give a ‘core’ genotype, there are genetic differences between the subtypes, which give rise to observed phenotypic differences (see [7], Chapter 2 for more details of the genetics of PWS). As well as two copies of all non-imprinted genes, people with the maternal disomy subclasses will have two copies of paternally imprinted (maternally expressed) genes, while those with the deletion subtype and the general population have a single copy that is expressed. It has been argued that this may have implications for our understanding of the high risk for psychosis in those with PWS due to maternal disomy [8]. Non-imprinted genes have two intact copies in general but if one copy is deleted, as in deletion subtypes, a recessive gene is more likely to be expressed. Longer deletions (Type 1 vs. Type 2) also result in greater impairment [9].

## 3. The Behavioural Phenotype of PWS

The PWS genotype gives rise to a ‘behavioural phenotype’ with particular cognitive, social and behavioural characteristics and a risk for behavioural disorders and psychiatric problems [7] (Chapters 6 and 7). Initially there is pronounced hypotonia and failure to thrive, followed by developmental and cognitive delay, preoccupation with food and hyperphagia, and relative growth and sex hormone deficiency (short stature and impaired sexual development). Emerging behavioural problems include temper outbursts, repetitive and ritualistic behaviours, mood swings, and skin picking [10]. We note that the distributions of these behaviours in PWS are similar in shape to those in the general population but the distributions in PWS are shifted towards the more severe end of the general population distributions. Inactivity is characteristic and leisure time is typically spent in solitary pursuits such as jigsaw puzzles, word search puzzles, television viewing, and computer games [11]. In the teenage years and early twenties, or later, depression and/or psychosis develop in some people. Rates of depression in deletion subtypes tend to be similar to those in other forms of intellectual disability but less in maternal disomy subtypes. However, in people with PWS due to maternal disomy, rates of psychosis are much higher (60%–100% lifetime prevalence) but not in those with a 15q11-q13 deletion. This is considered further later in the paper.

The most prevalent disorder in PWS is the preoccupation with food and propensity to overeat, which in one form or another affects everyone with the syndrome. Hyperphagia develops in early childhood following a period of failure to thrive. It has features in common with anorexia (high ghrelin, low growth hormone, and absent menses), with bulimia (binge eating), with addiction (food seeking and stealing) and with obsessive-compulsive disorder (OCD; constantly thinking about it). However, it is present from infancy and on the basis of behavioural and neuroimaging studies it is generally acknowledged to be a defect of satiety [12,13,14]. It might be predicted that a reasonably direct path between genotype and phenotype will eventually be established. As of yet, however, no loss of function of genes known to be involved in feeding pathways has been observed and the link between genotype and this particular phenotype has not been explained.

Repetitive and ritualistic behaviours in PWS have been interpreted as OCD by some authors and as part of the syndrome of autism by others [15,16]. However, most people with PWS do not have OCD (there is an absence of distress or resistance to the behaviours as would be expected with true OCD). Although they may also have social impairments, the majority of people with PWS do not meet the full criteria for autism [17,18,19,20,21] but they do exhibit behaviours (hoarding, the need to ask or tell and insistence on routine) similar to those of typically developing young children [22]. One paper suggests that these behaviours in PWS are best conceptualised as a consequence of arrested development [23] given their similarities with what is observed as part of normal childhood development. It is unlikely therefore that there is a direct link between genes and this specific behavioural profile, rather it is the impact of the genotype on brain development more generally that leads to the increased prevalence of such behaviours.

Temper outbursts have been conceptualised similarly and have also been linked to insistence on routine as well as deficits in executive functioning and task-switching. These outbursts are characteristically triggered by change or a refusal of some specific request and follow a course that is likely to include the rapid onset of verbal outbursts and sometimes physical aggression, eventually resolving with expressions of regret and distress [24,25,26]. This pattern is similar to what is observed in young children. Observations of marked reductions in the frequency and severity of such behaviours by the use of vagus nerve stimulation might indicate that the mechanism resulting in the high risk for such behaviours is the presence of a low threshold of the autonomic nervous system to respond in a rapid flight/fight mode when faced with an actual or perceived threat [27,28]. Thus, the mechanism is one of impaired emotional regulation, potentially for developmental reasons. Observational studies indicate that external contingencies clearly have an important role to play but fundamentally, the underlying mechanism that increases the risk for such behaviours is a shift downwards in the threshold for responding in this ‘fight’ manner, as opposed to seeking to engage and de-escalate the situation through social engagement.

Skin picking has been variously interpreted as self-harm, obsessive compulsive behaviour, or as a direct consequence of the loss of the Necdin gene since, in the case of the last of these, abnormal grooming behaviour has been observed in Necdin knockout mice. The observation that such behaviours usually occur where there has been an insect bite or some irritation suggests that local factors may initiate the behaviour, which may lead to a local inflammatory response and further irritation. A high pain threshold also means that there is a reduced negative consequence to skin picking. Factor analysis studies show that skin picking is not found on the same factor as repetitive and ritualistic behaviours suggesting that such behaviour is not primarily driven by obsessionality [23,29]. Hall et al. [30] undertook a functional analysis and reported that skin picking occurred more commonly under conditions of low attention, suggesting an interactive model between a biological vulnerability and environmental contingencies and setting conditions.

The most striking finding on psychiatric illness in PWS is the rate of psychotic illness in the maternal disomy genetic subclasses (including both mUPD and imprinting centre defects). This has been estimated at 60%–100% lifetime prevalence [31,32,33,34,35]. This suggests a strong genetic component in the cause of psychotic symptoms in PWS. Moreover, since the rate of psychosis in the deletion subtypes is no more than that in people with intellectual disabilities, psychosis is not due to having the core PWS genotype and the causal gene or genes would have to be either more strongly expressed from the maternal chromosome 15 or imprinted. The search for such a gene has been unsuccessful to date [36].

Soni et al. [31] suggested a two-hit model to account for the differential rates of psychosis in PWS. They hypothesized that the genetics of PWS results, regardless of genetic type, in an increased propensity to affective disturbance with the ‘second-hit’, having a chromosome 15 mUPD, resulting in the markedly increased risk for psychosis. This would suggest that the increased propensity of those with mUPD to develop psychosis must be a combination of both the effects of the PWS genotype on neural development and circuits in the brain that serve mood regulation, as well as gene dosage effects from maternally expressed genes on chromosome 15, which result in aberrant functioning of neural circuits and in the emergence of abnormal mental experiences [8,27].

### Deletion and mUPD Differences

Whilst this differential risk of psychotic illness must have a genetic basis this effect is likely to be mediated through differences in brain structure and function in those with the deletion compared with the maternal disomy form of PWS.

Cognitive function and brain structure have been shown to differ between PWS genetic subtypes, indicating potential differences in developmental pathways. The question remains as to whether such differences might in turn explain the differential risk for psychotic illness. PWS, irrespective of genetic type, is associated with mild to moderate intellectual disability, but differences in cognitive profiles between the subtypes have been identified. The studies typically report a full-scale IQ (FSIQ) relatively similar between genetic subtypes, but a slightly higher performance IQ (PIQ) in those with PWS due to a deletion, and a higher verbal IQ (VIQ) score but a greater impairment in processing speed in those with mUPD [37]. FSIQ seems more homogenous in those with deletion PWS with similar VIQ and PIQ, whereas those with mUPD have a higher average VIQ than PIQ [37,38,39]. Whittington et al. (2004) compared subtests scores across PWS genetic subtypes and a comparison mixed aetiology intellectual disability group [37]. The cognitive profile of the mUPD group differs from the deletion PWS and LD group, who have very similar profiles (Figure 2).

From a neurophysiological angle, an encephalogram (EEG) study using a Go/No-Go task [40], found a reduced sensory processing speed in people with mUPD compared to those with deletion PWS. People with mUPD were reported to have significantly increased reaction times compared to those with deletion PWS and healthy controls, with deficits in both the N200 and P300 peaks related to early modality-specific inhibition and late general inhibition, respectively. Those with deletion PWS showed impairment only for N200 modulation. Another PWS study found a specific deficit in segregating human voices from a noisy background, and a failure to fully process sensory information before initiation of a behavioural response in PWS, again the deficit being greater in those with mUPD compared to those with deletion PWS [41]. A recent study in children with PWS has found significantly reduced white matter microstructure in most of the major white matter tracts in people with PWS due to mUPD compared to deletion PWS, similar to those reported in people with schizophrenia or those at ultra-high risk for psychosis [42,43]. The latter two populations have also been found to have an impaired processing speed [37,40,44]. Freedman et al. [45] observed compromised P50 sensory gating in patients with schizophrenia as well as some of their unaffected relatives, suggesting a role for such impairments in the aetiology of psychosis. Thus, impairments in both auditory processing and processing speed have been found in people with schizophrenia, those at risk for psychosis, and in people with PWS due to mUPD, suggesting at least partially common brain mechanisms and potential targets for treatment development.

## 4. Genetically Determined Neurodevelopmental Syndromes

Among the population of people with ID are groups with various genetically determined neurodevelopmental syndromes (Angelman, Cornelia de Lange, Cri-du-chat, Down, Fragile-X, Lesch Nyhan, Prader-Willi, Rett, Smith-Magenis, Tuberous sclerosis complex, Velocardiofacial, Williams, etc.). Individuals with these genetic syndromes have distinct phenotypes, including distinctive patterns of adaptive and maladaptive behaviours that distinguish one syndrome from others—referred to as the ‘behavioural phenotype’ of the syndrome [46]. In these cases the phenotypic behaviours and the characteristic phenomenology of the underlying mental state of the person may make up symptom clusters that then meet diagnostic criteria for a specific mental disorder, leading to a diagnosis, for example, of an anxiety disorder in an individual with Fragile-X or Williams syndrome [47]. In both of these syndromes ‘anxiety’ in different ways is common and the propensity to anxiety is considered to be characteristic of people with these syndromes. However, what is meant by saying that someone with one or other of these syndromes is ‘anxious’ and whether the underlying brain mechanisms between syndromes are similar or not (and therefore treatments may be similar) is far from clear but phenomenologically they appear to be different. Observed behaviours, such as self-injurious behaviour, may also be helpful diagnostically. For example, in the case of people with Lesch Nyhan syndrome (LNS) such behaviour is a key part of the phenotype of that syndrome. Its presence during childhood in males might well alert one to the diagnosis of this syndrome. In addition, the early onset, severity, and extent of this behaviour indicate that there is a direct link between the syndrome genotype and that aspect of the phenotype. In LNS the behaviour is universal, of early onset in life, and does not fit the criteria for other established mental disorders. This example highlights the importance to our understanding of the behavioural and mental health aspects of children and adults with neurodevelopmental syndromes to distinguish between: (1) whether a particular symptom (e.g., self-injurious behaviour) is a diagnostic feature of the syndrome itself; or (2) whether it is best understood as either a symptom of some additional co-morbid mental disorder; or (3) it is best understood through a different conceptual framework, such as that of the operant model of applied behavioural analysis. In this latter case the focus is on identifying external factors that predispose to, precipitate or are maintaining specific behaviours, or are internal or external setting conditions for such behaviours. As is considered in the case of PWS this is an important step in seeking to explore the relations between the syndrome genotype and its phenotype.

### 4.1. Diagnostic Criteria

Biomedical systems of classification, such as the DSM-5 and ICD-10, have done much to bring rigor into our definitions and understanding of mental disorder, a very necessary process if specific treatments are to be developed and tested. DSM-5 proposes that each of the mental disorders is defined by an agreed of symptoms or behaviours, not all of which are necessary and none of which is sufficient for a definite diagnosis (because they can appear in more than one such list). Characterised as a clinically significant behavioural or psychological syndrome or pattern that occurs in an individual that is associated with present distress or disability or with a significantly increased risk of death, pain, disability, or an important loss of freedom. This syndrome or pattern must not be merely an expectable and culturally sanctioned response to a particular event, such as a bereavement.

Mental disorder and mental illness may be difficult to define and diagnose and judgements have to be made as to what is clinically significant or an expectable culturally sanctioned event. There is no single definitive test for anxiety, depression, psychosis, bipolar disorder, or obsessive-compulsive disorder and there is an overlap of symptoms between diagnostic categories in the standard instruments DSM-5 and ICD 10. There is also a debate as to whether particular forms of mental disorder, such as autism spectrum conditions, are best considered as discrete (yes/no) or a continuum where a line divides those considered to have a positive diagnosis and are therefore ‘atypical’ from those considered to be ‘typical’. This is similar to specific physical disorders, such as type 2 diabetes and hypertension—the recommendations as to where the cut off lies, above which some (kind of) treatment is recommended, changes over time. These difficulties are further compounded, in the case of people with intellectual disabilities; firstly because of the complexity associated with an atypical pattern of development; secondly, because the individuals concerned may have difficulty in communicating feelings and symptoms; and thirdly their level of functioning and behaviour may not conform to general expectations for their age. However, despite the limitations of this approach, the diagnosis of certain mental disorders, such as the major mental illnesses of bipolar disorder and schizophrenia, have, through the presence of a recognised grouping of specific characteristics come to imply some understanding as to likely causation, and this informs treatment and prognosis.

Prader-Willi syndrome, like other genetically determined neurodevelopmental syndromes, is considered to fall under the broad umbrella term of ‘mental disorder’. PWS may be additionally classified as ‘intellectual developmental disorder’ if the necessary criteria are met. As we summarized above, PWS has an early clinical profile and developmental history that indicates involvement of different organ systems. These include the central nervous system, resulting in the presence of functional impairments and disabilities. In addition, for a significant proportion of children and adults with PWS, patterns of behaviour and particular mental states (the ‘behavioural phenotype’ of PWS) can be observed during their lifetime that also have a major impact on functioning. Some of these meet established diagnostic criteria for a co-morbid mental disorder and others do not. Psychosis (mainly in the disomy subtype) and depressive illness (mainly in the deletion subtype) do meet criteria, and this informs treatment by guiding as to the medications that might be tried. The question then arises as to how the additional behavioural phenomena are best conceptualised, characterised, and understood so that ultimately effective interventions are developed.

### 4.2. Applied Behavior Analysis

In contrast, another term, which is common in the intellectual disabilities literature, is that of ‘challenging behaviour’, attributed to Emerson [48] and defined as follows:
“Culturally abnormal behaviour(s) of such an intensity, frequency or duration that the physical safety of the person or others is likely to be placed in serious jeopardy, or behaviour which is likely to seriously limit use of, or result in the person being denied access to, ordinary community facilities.”

Individual descriptions of challenging behaviour do not mimic the diagnostic process by attempting to group together specific signs or symptoms that characterises the behaviour in a manner that informs an understanding of causation or treatment. Whilst the presence of such behaviours may be an indication of some underlying and as yet undiagnosed mental disorder, the use of the term ‘challenging behaviour’ does not imply that the criteria for a specific mental disorder are met. The term is descriptive and is a marker for the level of impact of the behaviour and an indication that those providing support should seek to accommodate to the behaviours. In contrast to mental disorder, our more recent understanding of challenging behaviour, certainly as applied to people with intellectual disabilities, has been primarily led by the theoretical constructs of ABA. In this view, it is argued that such behaviours are under operant control and have come to have specific functions, such as avoiding demands or maintaining attention. In this context there has been a clear resistance to defining such behaviour as ‘a mental disorder’. There remains an on-going tension between the biological and the psychological: what might be innate, what might be a disorder of cerebral function, what is learnt, and/or what is a consequence of factors in the physical and emotional environments that have shaped and reinforced these behaviours [49]. These and other conceptual issues are complex enough when applied to the typically developing population but can be even more problematic when it comes to children and adults with neurodevelopmental disorders. However, a failure to address some of these issues, as they apply to our understanding of mental health and behaviour disorders in children and adults with neurodevelopmental disorders, such as PWS, hinders research, impairs the development and trials of new and innovative treatments, and limits our ability to track a course from genotype to behavioural phenotype.

### 4.3. Research Domain Criteria (RDC)

More recently a different perspective and conceptual framework, that of Research Domain Criteria (RDC), has shifted the emphasis away from diagnoses using the DSM-5 system to linking psychopathology to discoveries in genetics and neuroscience. In this conceptual framework behavioural and psychiatric disorders are consider to arise as a consequence of disturbances in brain networks [50]. The system proposes five ‘domains’ to classify behaviour: negative valence, positive valence, cognitive processes, social processes, and arousal/regulatory systems. Using non-suicidal self-injurious behaviour as an example, it has been argued that diagnostic approaches fail as clinical characterisation does not map directly onto brain mechanisms whereas Research Domain Criteria seek to do that—whether they do is a matter for empirical study [51,52]. These different approaches, illustrated in Figure 3, each have their strength and utility.

The main strength of the RDC is that it seeks to classify by underlying brain mechanisms, rather than by lists of symptoms. In doing so it brings together psychological constructs, some of which are familiar to the ABA approach (reinforcement, reward, etc.); what is known about the neural basis of specific behaviours relevant to our understanding of PWS, such as hyperphagia (e.g., feeding pathways projecting from the hypothalamus to frontal areas of the brain); and specific neural circuits that underpin particular cognitions, such as executive functioning, the disruption of which may result in impairments in such tasks as attention shifting, planning, etc. This is particularly relevant to genetic syndromes, since a behaviour that is prevalent in and peculiar to a given syndrome must be directly or indirectly (for example, by its effect on regional brain development) connected to the genetics of that syndrome. Relevant to this approach, we note that in PWS functional proton emission tomography (fPET) and functional magnetic resonance imaging (fMRI) have been used to study the hyperphagia [14], attention switching as a trigger for temper outbursts [25], and skin picking behaviour [30].

## 5. Conclusions

The purpose behind establishing a diagnosis and of the diagnostic process, in general, is to arrive at an understanding of a particular syndrome, abnormal mental state, or behaviour. By accurately defining these different conditions through history taking, observation, and investigation, it is possible to compare with others who may have apparently similar conditions and to investigate causative mechanisms and, ultimately, to develop and test treatments. We propose that this fundamental proposition is still an appropriate clinical approach when it comes to neurodevelopmental syndromes, such as PWS, but that it must be more nuanced and involve different layers and models of understanding. The Research Domain Criteria takes this further and offers a different perspective seeking to link genes, brain, and behaviour [52]. This approach, in attempting to map behaviour to brain function, proposes using various neuroscience techniques to investigate the negative and positive valence of specific behaviours, the involvement of reward and other brain circuits, the role of social and sensory processing and underlying brain networks, and of the arousal systems of the brain. In order to develop our understanding of challenging behaviour, the need is to investigate whether, within and between groups of children and adults with different neurodevelopmental syndromes, specific aspects of the behaviour can be seen to cluster together and whether such behaviour can be mapped to dysfunction in particular neural systems, such as the reward circuits. What then are the key characteristics in the history, when observing challenging behaviour, that are important and which might then guide our understanding of mechanisms and ultimately of treatment?

We propose that:

Firstly, research needs to focus on the phenomenology of behaviours and mental states that are observed in those with neurodevelopmental syndromes. In the case of PWS this includes the hyperphagia, repetitive and ritualistic behaviours, temper outbursts, skin picking, and abnormal mental states. In order to move closer to understanding the causative mechanism and to establishing whether superficially similar behaviours across neurodevelopmental syndromes are likely to be similar or different, these behaviours and abnormal mental states need to be described in two major domains. One is the age of onset and course during development and over the lifetime. The other is an accurate description of the phenomenology. In the case of hyperphagia in PWS, for example, it includes the observations that the onset is in early childhood and persists probably throughout life, the severity of the behaviour and its consequences, the presence of food pre-occupations, food stealing, hoarding, etc. Similarly, with skin picking it would include age of onset, topography, course and phenomenology, and whether self-restraint is a feature. Detailed description would allow accurate comparisons of apparently similar phenomena within and across neurodevelopmental syndromes and in the typically developing population.

Secondly, the nature and clustering of the phenomenology and the observed behaviours and their course through time should be examined and the question asked as to whether or not they meet, or approximate to, established criteria for a known mental disorder. In the case of the repetitive and ritualistic behaviours in PWS this would clearly indicate that they are different from those observed in OCD, however, they are similar, but not identical, to those observed in people with autism spectrum disorders [19]. Thus, as a working hypothesis such behaviours may be best considered as of developmental origin rather than acquired during life. In contrast, the abnormal mental states that may appear in the teenage years or early adult life, particularly, in those with the mUPD form of PWS, may reasonably be called a psychotic illness even if it cannot be more narrowly defined as schizophrenia or bipolar disorder [53].

Thirdly, these behaviours or observed abnormalities of mental state should be considered from the perspective of causation. The question to be addressed is in essence which conceptual model of understanding best accounts for what is observed (phenomenology) and for the age of onset and the course over the lifespan. The models proposed are not mutually exclusive but we suggest that it is this process that is critical to understanding and which leads to informed intervention. Syndrome specific solutions may be required.

However, the approach put forward in the Research Domain Criteria is to ask the question: can particular behaviours, such as hyperphagia, be mapped against known neural networks and connections? By doing so, specific additional features in the history or on observation might be found to clump together and might be considered to have a common underlying mechanism and respond to a specific treatment. Such an approach will be informed by more detailed study using techniques such as neuroimaging. In PWS fPET and fMRI have been used to study the hyperphagia [14], attention switching as a trigger for temper outbursts [25], and skin picking behaviour [30]. This approach investigates the cerebral signature (endophenotype) that specifically underpins the behaviour in question and may well be able to determine whether similar behaviours across people with different syndromes have similar mechanisms. Whether similar behaviours but clustering together with different clinical features (e.g., self injurious behaviours with different topographies or with or without self restraint) have different mechanisms. In PWS, the hyperphagia may be best considered as a developmental abnormality characteristic of PWS that maps onto the functioning of the nuclei in the hypothalamus, and the connections and projections between the hypothalamus and the cortex that map onto areas of the brain that represent the conscious experiences of hunger and fullness. The repetitive and ritualistic behaviours and temper outbursts may be seen as a consequence of a more general abnormality of development resulting in poor emotional control and a low threshold for outbursts at minor real or perceived challenges that map onto the central connections of the autonomic nervous system and the limbic system. The abnormalities of the mental state, particularly observed in people with mUPD, which are a consequence of the development of a co-morbid mental illness might map to particular neural transmitter networks in the brain. For example, sensory processing is slower in those with mUPD, as it is in people at high risk for schizophrenia [8] and levels of cerebral GABA have been found, using magnetic resonance imaging (MRI), to be reduced in people with PWS [54]. One neuroimaging study has mapped skin picking to those areas of the brain (R insula and L precentral gyrus) mediating introceptive behaviours (itch and pain) [30].

In summary, we proposed that the diagnostic approach still has value in the identification of specific co-morbidities that may present with behavioural changes. In addition, we suggest that the Research Domain Criteria approach lends itself to the study of those behaviours, which severely impact on a person’s life, where these behaviours are at present not easily categorised. Through an approach similar to that of the diagnostic process, these behaviours should be classified on the basis of their characteristics. This in turn may result in the identification of aetiologically distinct sub-groups of superficially similar behaviours. These may map to different neural networks and by implication will require different treatment approaches. The long-term aim is to be able to identify where specific abnormalities of gene expression impact on brain development and functioning in a manner that results in a recognisable and characteristic pattern—the behavioural phenotype of that syndrome.

## Figures and Tables

**Figure 1 genes-10-01025-f001:**
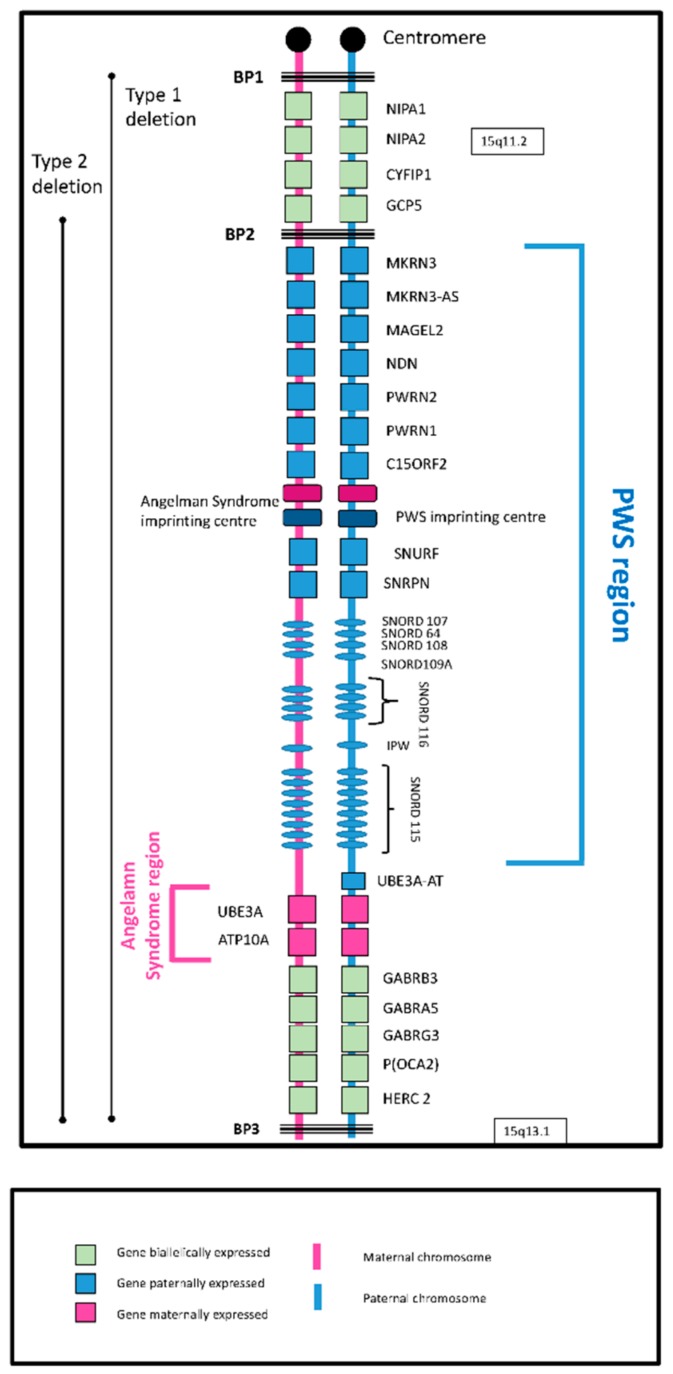
The genetics of Prader-Willi syndrome (PWS).

**Figure 2 genes-10-01025-f002:**
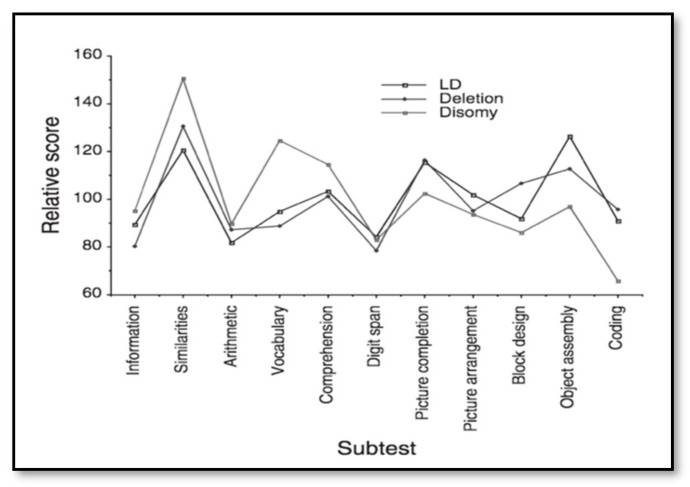
Weschler subtest profiles of mUPD, deletion PWS (delPWS), and learning disability groups from Whittington et al. 2004 [37].

**Figure 3 genes-10-01025-f003:**
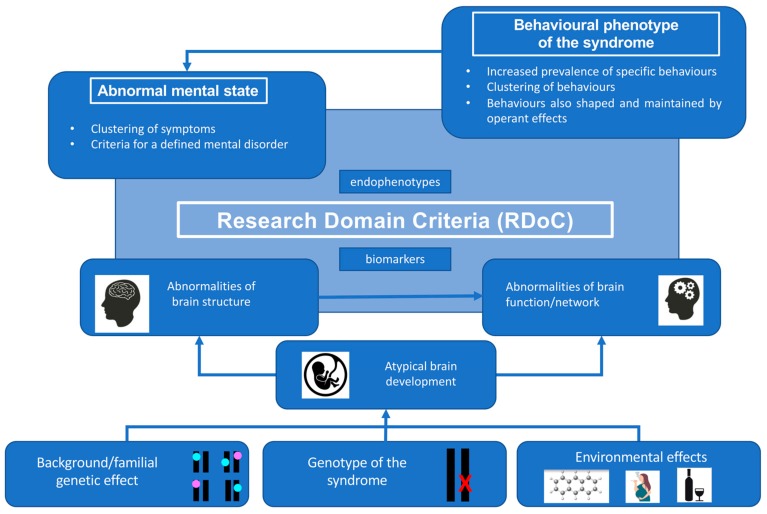
The genotype of the syndrome, as well as the background genetic and environmental effects, cause abnormalities in brain development, leading to abnormalities in brain structure and/or function. The top two boxes correspond to the diagnostic and applied behavioural analysis (ABA) approaches and the pale blue box to the Research Domain Criteria (RDC) approach.

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
