# Peer review of "Defining Mental and Behavioural Disorders in Genetically Determined Neurodevelopmental Syndromes with Particular Reference to Prader-Willi Syndrome"

_genes, 2019, doi:10.3390/genes10121025_

Round 1

Reviewer 1 Report

This is a conceptually dense paper that purports to “propose a taxonomy for psychiatric and behavioural disorders in genetically determined neurodevelopmental syndromes that brings together diagnostic, psychological and developmental approaches”.  While the paper falls far short of these lofty claims, the proposed diagnostic approach and extensively delineated example for understanding  syndrome specific disordered behavior is thought provoking.  And, if substantially edited and tightened up could serve as a nice foundational  overview and framework for those interested in syndrome specific mental and behavior disorders, particularly Prader-Willi syndrome.

One difficulty with reading this paper is the style of writing. It is extremely word dense, contains long complex sentences requiring several readings to understand what the sentence is saying,  and much of the writing is in the passive voice. Such complexity frequently masks or impedes understanding the point being made. For instance: the second paragraph of the introduction starts:

Prader-Willi Syndrome (PWS), in the context of the taxonomy of mental disorders as set out in manuals such as DSM-V[1] and ICD-10[2], like other genetically determined neurodevelopmental syndromes, is considered to fall under the broad umbrella term of ‘mental disorder’, and, if the  necessary criteria are met, also under the category of ‘intellectual developmental disorder’.

Simple editing would make it would be so much easier to follow: Prader-Willi syndrome (PWS), like other genetically determined neurodevelopmental syndromes, falls under the umbrella term of  “mental disorder” in the DSM-5 and ICD-10 taxonomies. And, PWS may be additionally classified as an “intellectual developmental disorder” if the necessary criteria are met.   

Some substantive concerns:

In the section titled Diagnostic Criteria and other definitions, the point is made that rather than diagnosing mental disorders among those with intellectual disabilities based on the discrete disorders defined in DSM-5 and ICD-10, an alternative description using the term “challenging behavior” is employed. The authors suggest that this descriptive terminology connotes sufficiently severe problem behaviors  that  “services” may be required yet avoids inappropritely labeling someone as “mentally disordered”.  Finally the authors assert “our understanding of challenging behaviors has been primarily led by the theoretical constructs of applied behavior analysis and it is for that reason that there has been a resistance to defining these behaviors as “a mental disorder””.  I suggest that even a casual review of the literature finds the use of the term challenging behavior utilized long before the relatively recent resurrection of applied behavior analysis intervention techniques. It was used to connote a problem behavior that may need treatment but is not “diagnosable” with prevailing mental disorder taxonomies and, in  fact, is not  a “mental disorder” in the traditional sense. The authors themselves illustrate such a case when describing the self-injurious behavior inherent in Lesch-Nyhan syndrome. As the authors state, “the behavior is universal in those with the syndrome…and does not fit criteria for other established mental disorders”.  

The authors want to build the argument that the dysfunctional neural circuitry  approach of Research Domain Criteria offers a more parsimonious framework for understanding mental disorders in genetically determined neurodevelopmental disorders using Prader-Willi syndrome as an example. Thus, I would suggest that the logic behind the argument would be better explicated if the order of the two sections “genetically determined neurodevelopmental syndromes” and  “Research Domain Criteria” were reversed.

In the section titled Prader-Willi syndrome, in the paragraph beginning “although the underlying cause gives……the second sentence indicates “Disomy subtypes will have two copies of paternally imprinted genes…” I believe that is a typo as the authors have previously defined the loss of expression based on maternally imprinted and paternally expressed. Although Dad is certainly silenced in the case of the disomy, it is not accurate to term that paternally imprinted.

In the paragraph starting “repetitive and ritualistic behaviors, the authors that the majority of individuals with PWS do not have autism – presumably based on the presence of repetitive and ritualistic behaviors. They fail to incorporate the extreme social deficits and language processing difficulties that play into the diagnosis of autism spectrum disorders (under DSM-5 language is removed as a criteria). Both social deficits and language processing concerns are empirically noted to be ubiquitous in this population to the point that most affected individuals meet criteria for autism spectrum disorder.

 While the authors are attempting to explicate a view of PWS through the lens of RDoC, throughout the paper there appears to be a subtle agenda to similarly frame the behavioral explanations from the authors theory of arrested development. If we are to refocus our view of the behaviors inherent in genetically driven neurodevelopmental syndromes, and in this instance PWS, this must remain an empirical question.

Author Response

We acknowledge the complexity of some of the sentences in the submitted article.  We have reframed the worst of these, either through splitting into shorter sentences or through punctuation. We have also re-written parts of the paper to add clarity to the arguments we are proposing and address the issue of splitting and clumping symptoms..

We have included the definition of challenging behaviours as first proposed by Emerson and clarified the fact that such behaviour does not fit criteria for mental disorder and how applied behavioural analysis has come to be a significant conceptual framework for understanding such behaviour.

We agree that the order of sections ‘genetically determined neurodevelopmental syndromes’ and ‘Research domain criteria’ should be reversed. This has been done.

Disomy subtypes will have two copies of paternally imprinted (maternally expressed) genes because they have two intact maternally marked chromosome 15s.  This is correct as written.

We state that the majority of people with PWS do not have autism, and cite evidence from research papers that report differences between groups with PWS and groups with diagnosed autism.  In particular Dykens et al. (2017)  found that the diagnosis of autism could be made in

only 12.3% of their sample. We have acknowledged the PWS social difficulties.  This reference has been added.

‘Arrested development’ was mentioned only once - in the context of the prevalence in PWS of ‘hoarding, the need to ask or tell and insistence on routine’ similar to the behaviours of typically developing young children  The persistence of these behaviours throughout life in people with PWS does look like arrested development.  We do not understand the referee seeing it ‘ throughout the paper’.

Reviewer 2 Report

Holland, Aman and Whittington discuss the interplay between genotype, brain molecular and structural phenotype, clinical and behavioral phenotype in the context of neurodevelopmental disorders. They use Prader-Willi syndrome as an example. While everything they state and write is correct, I do not see the novelty of their approach and manuscript. Their key message is the following: in order to understand the clinical, behavioral and psychiatric phenotype of an individual, one needs to perform thorough phenotyping. This includes a detailed history, physical exam, and investigation of the underlying causes. Different genetic etiologies may account for different nuances of clinical phenotype, even within the same disease entity. This is not a new concept. A key question behind, not really discussed in the manuscript, is whether to lump or split disease entities, and how etiological groups or clusters can be defined, in order to devise common treatment approaches, both clinically, but also for the sake of clinical trials.

Author Response

We are pleased that the referee finds that ‘everything they state and write is correct’.  With regard to novelty, no references are provided by the referee to similar work and we have not been able to find  similar work on PWS.

We have expanded discussion relating to challenging behaviour. We thank the referees for their comments, particularly the very helpful comments of referee #1

Round 2

Reviewer 1 Report

The authors have certainly done a significant edit of the paper, and in many areas it is tighter, while in others it has lost focus and impact. From a word dense perspective it is markedly improved. However, I note particularly in the abstract and introduction, the goal of shifting the diagnostic framework from DSM-5 and ICD10 to a Research Domain Criteria is not indicated leaving the reader with an incomplete map regarding why they might want to continue reading the paper.

As I understand the goals of the paper, the authors seek to indicate that usual psychiatric diagnostic frameworks often fail when applied to an intellectually disabled population (with or without a genetic origin) and thus the term “challenging behavior” has been employed to indicate significant functional abnormalities needing treatment but are inadequately categorized by the usual diagnostic schema – an analysis that few in the field would find new or arguable.  They then want to propose that the RoDC is a new or perhaps unifying framework to use in an ID population which they seek to exemplify with PWS.  However, in this version we essentially have a brief critique up front  of DSM-5 and ICD#10, then a good overview of PWS, and then a continuing extensive critique of current schema in the conclusion, but no real integration of how RofDC would move anything forward.

Most readers will have no argument with the concept that a deeper dive into the weeds of neuro/psych/phys functioning is necessary when understanding the “challenging” behavior among those with ID and particularly those of genetic origin. And, the RofDC framework may be one way of conceptualizing that search through the weeds. However, that integration is still lacking in this version of the paper. I hope the authors will try again.  

Author Response

We agree with the referee that the main argument of the paper is not clear in the introduction and abstract.  We have added to the abstract and added a first paragraph to the introduction to clarify the thread running through the narrative.

Reviewer 2 Report

The manuscript has improved. Needs editorial editing (multiple occasions of words repeating themselves). 

Author Response

We have removed the typos.